



**Quantifying streamflow and active groundwater storage in response to climate**
**warming in an alpine catchment, upper Lhasa River**
Lu Lin[a,b], Man Gao[c], Jintao Liu[a,b*], Jiarong Wang[a,b], Shuhong Wang[a,b*], Xi Chen[a,b,c],
Hu Liu[d]
[a] *State Key Laboratory of Hydrology-Water Resources and Hydraulic Engineering,*
*Hohai University, Nanjing 210098, People's Republic of China*
[b] *College of Hydrology and Water Resources, Hohai University, Nanjing 210098,*
*People's Republic of China*
[c] *Institute of Surface-Earth System Science, Tianjin University, Tianjin 300072,*
*People's Republic of China*
[d] *Linze Inland River Basin Research Station, Chinese Ecosystem Research Network,*
*Lanzhou 730000, People's Republic of China*
*\* Corresponding author. Tel.: +86-025-83787803; Fax: +86-025-83786606.*
*E-mail address: jtliu@hhu.edu.cn (J.T. Liu).*



**Abstract**
Climate warming is changing streamflow regimes and groundwater storage in cold
alpine regions. In this study, a headwater catchment named Yangbajain in the Lhasa
River Basin is adopted as the study area for quantifying streamflow changes and
active groundwater storage in response to climate warming. The changes in
streamflow regimes and climate factors are evaluated based on hydro-meteorological
observations from 1979 to 2013. The results show that annual streamflow increases
significantly at a rate of about 12.30 mm/10a during this period. Through baseflow
recession analysis, we also find that the estimated groundwater storage that is
comparable with the GRACE data increases significantly at the rates of about 19.32
mm/10a during these years. The rising of air temperature is the main factor for the
increase in streamflow and groundwater storage, which has led to a loss of over 25%
of the total glacier volume for half century in this catchment. Parallel comparisons
with other sub-basins in the Lhasa River Basin reveal that the increased streamflow at
the Yangbajain station is mainly fed by the accelerated glacier retreat rather than
frozen ground degradation. However, the increase of active storage capacity is caused
by frozen ground degradation, which can accommodate the increasing meltwater in
the valley. The huge gap between the melt-derived runoff and the increased water
volume in groundwater storage and streamflow suggests that more than 60% of the
total ablation of glaciers should be discharged downstream through deep fault. This
study provides a perspective to clarify the impact of glacial retreat and frozen ground



degradation on hydrological processes, which fundamentally affects the water supply
and the mechanisms of streamflow generation and change.
**Keywords:** Climate warming; Streamflow; Groundwater storage; Glacier retreat;
Frozen ground degradation; Tibetan Plateau



## 1. Introduction

Often referred to as the "Water Tower of Asia", the Tibetan Plateau (TP) is the
source area of major rivers in Asia, e.g., the Yellow, Yangtze, Mekong, Salween, Indus,
and Brahmaputra Rivers (Cuo et al., 2014). The delayed release of water resources on
the TP through glacier melt can augment river runoff during dry periods, giving it a
pivotal role for water supply for downstream populations, agriculture and industries in
these rivers (Viviroli et al., 2007; Pritchard, 2017). However, the TP is experiencing a
significant warming period during the last half century (Kang et al., 2010; Liu and
Chen, 2000). Along with the rising temperature, major warming-induced changes
have occurred over the TP, such as glacier retreat (Yao et al., 2004; Yao et al., 2007)
and frozen ground degradation (Wu and Zhang, 2008). Hence, it is of great
importance to elucidate how climate warming influences hydrological processes and
water resources on the TP.
In cold alpine catchments, a glacier is known as a "solid reservoir" that supplies
water as streamflow, while frozen ground, especially permafrost, servers as an
impermeable barrier to the interaction between surface water and groundwater
(Immerzeel et al., 2010; Walvoord and Kurylyk, 2016). Since the 1990s, most glaciers
across the TP have retreated rapidly due to global warming and caused an increase of
more than 5.5% in river runoff from the plateau (Yao et al., 2007). Meltwater is the
key contributor to streamflow increase especially for headwater catchments with
larger glacier coverage (>5%) (Bibi et al., 2018). Meanwhile, in a warming climate,



numerous studies suggested that frozen ground on the TP has experienced a noticeable
degradation during the past decades (Cheng and Wu, 2007; Wu and Zhang, 2008).
Frozen ground degradation can modify surface conditions and change thawed active
layer storage capacity in the alpine catchments (Niu et al., 2011). Thawing of frozen
ground increases surface water infiltration, supports deeper groundwater flow paths,
and then enlarges groundwater storage, which is expected to have a profound effect
on flow regimes (Kooi et al., 2009; Bense et al., 2012; Walvoord and Striegl, 2007;
Woo et al., 2008; Ge et al., 2011; Walvoord and Kurylyk, 2016).
It is challenging to understand how glacier melt and frozen ground thaw alters the
mechanism of streamflow in a warmer climate due to the complicated interactions
between hydrological and cryospheric processes. In earlier phase of glacier melt,
accelerated glacier retreat will bring large quantities of meltwater available directly
for surface runoff or indirectly for groundwater recharge (Bayard et al., 2005).
Meanwhile, frozen ground thawing may allow for increased groundwater recharge
from meltwater infiltration (Evans and Ge, 2017). Generally, climate warming is
hypothesized to generate a quantitative and temporal shift in the partitioning of
meltwater between surface runoff and groundwater flow, and thereby alter the
quantity and timing of baseflow (Green et al., 2011; Evans et al., 2018). Through
groundwater modeling, Evans et al. (2015) found an increase in mean annual surface
temperature of 2 °C reduced approximately 28% of the areal extent of permafrost and
tripled baseflow contribution to streamflow in a headwater catchment on the northern



TP. Qin et al. (2016) discovered that the increasing precipitation and the thawing of
frozen ground were the main factors on the increase of baseflow with no significant
change in surface runoff in the upper Heihe River Basin of the northeastern TP.
Previous data-based studies indicated that the baseflow has increased especially
during winter with a reduction or no pervasive change in summer streamflow in the
central and northern TP (Liu et al., 2011; Niu et al., 2016) as well as Arctic rivers
(Walvoord and Striegl, 2007; Smith et al., 2007; St. Jacques and Sauchyn, 2009).
Moreover, based on numerical simulations, Bense et al. (2012) suggested that the
increasing groundwater storage caused by frozen ground degradation would delay
baseflow increase possibly by several decades to centuries. A slowdown in baseflow
recession was found in the northeastern and central TP (Niu et al., 2011; Niu et al.,
2016; Wang et al., 2017), in northeastern China (Duan et al., 2017), and in Arctic
rivers (Lyon et al., 2009; Lyon and Destouni, 2010; Walvoord and Kurylyk, 2016).
While, previous studies were important for understanding the effects of climate
warming on hydrological changes in cold alpine catchments (Niu et al., 2011; Niu et
al., 2016; Wang et al., 2017). However, quantitatively characterizing storage
properties and sensitivity to climate warming in cold alpine catchments is still
important for local water as well as downstream water management (Staudinger,
2017). Moreover, revealing the storage characteristics makes it easier to predict
hydrological cycle and streamflow changes response to a warming climate in cold
alpine catchments (Singleton and Moran, 2010). Thus, this study focuses on



quantifying streamflow and aquifer storage volume response to changes in glacier
melt and frozen ground thaw at the catchment scale on the southern TP. However, it is
difficult to directly measure catchment aquifer storage (Staudinger, 2017; Käser and
Hunkeler, 2016) and the GRACE data has low resolution and accuracy in assessing
total groundwater storage changes at the catchment scale (Green et al., 2011). An
alternative method, namely, recession flow analysis, can theoretically be used to
derive the active groundwater storage volume to reflect frozen ground degradation in
a catchment (Brutsaert and Nieber, 1977; Brutsaert, 2008). For example, the
groundwater storage changes can be inferred by recession flow analysis assuming
linearized outflow from aquifers into streams (Lin and Yeh, 2017). Due to the
complex structures and properties of catchment aquifers, the linear reservoir model
may not sufficient to represent the actual storage dynamics (Wittenberg, 1999;
Chapman, 1999; Liu et al., 2016). Hence, Lyon et al. (2009) adopted the nonlinear
reservoir to fit baseflow recession curves for the derivation of aquifer attributes,
which can be developed for inferring aquifer storage. Buttle (2017) used Kirchner's
(2009) approach for estimating the dynamic storage in different basins and found that
the storage and release of dynamic storage may mediate baseflow response to
temporal changes.

In this study, the Yangbajain Catchment in the Lhasa River Basin is adopted as the

study area. The catchment is experiencing glacier retreat and frozen ground
degradation in response to climate warming. The main objectives of this study are (1)





to quantify the changes between surface runoff and baseflow in a warming climate; (2)
to quantify active groundwater storage volume by recession flow analysis; (3) to
analyze the impacts of the changes in active groundwater storage on streamflow
variation. The paper is structured as follows. The section of Materials and Methods
includes the study area, data sources and methods. The Results and Discussion
sections present the changes in streamflow and its components, climate factors, and
glaciers, and we will discuss the changes in streamflow volume and baseflow
recession in response to the changes in active groundwater storage. The main
conclusions are summarized in the Conclusions section.
**2. Materials and Methods**
**2.1. Study area**
The 2,645 km$^2$ Yangbajain Catchment in the western part of the Lhasa River Basin
(Figure 1a) lies between the Nyainqêntanglha Range to the northwest and the
Yarlu-Zangbo suture to the south. In the central of the catchment, a wide and flat
valley (Figure 1b) with low-lying terrain and thicker aquifers is in a half-graben
fault-depression basin caused by the Damxung-Yangbajain Fault (Wu and Zhao, 2006;
Yang et al., 2017). As a half graben system, the north-south trending
Damxung-Yangbajain Fault (Figure 1b) provides the access for groundwater flow as
manifested by the widespread distribution of hot springs (Jiang et al., 2016). The
surface of the valley is blanketed by Holocene-aged colluvium, filled with the great
thickness of alluvial-pluvial sediments from the south such as gravel, sandy loam, and





clay. The vegetation in the catchment is characteristic of alpine meadow, alpine steppe,
marsh, shrub, etc; meadow and marsh are mainly distributed in the valley and river
source (Zhang et al., 2010).

Located on the south-central TP, the Yangbajain Catchment is a glacier-fed

headwater catchment with significant frozen ground coverage (Figures 1b & 1c). A
majority of glaciers were found along the Nyainqêntanglha Ranges (Figure 1b).
Glaciers cover over ten percent of the whole catchment, making it the most
glacierized sub-basin in the Lhasa River Basin. According to the First Chinese Glacier
Inventory (Mi et al., 2002), the total glacier area was about 316.31 km$^2$ in 1960. The
ablation period of the glaciers ranges from June to September with the glacier termini
at about 5,200 m (Liu et al., 2011). According to the new map of permafrost
distribution on the TP (Zou et al., 2017), the valley is underlain by seasonally frozen
ground (Figure 1c). It is estimated that seasonally frozen ground and permafrost
accounts for about 64% and 36% of the total catchment area, respectively (Zou et al.,
2017). The lower limit of alpine permafrost is around 4,800 m, and the thickness of
permafrost varies from 5 m to 100 m (Zhou et al., 2000).

The catchment is characterized by a semi-arid temperate monsoon climate. The

average annual air temperature of the Yangbajain Catchment is approximately -2.3 ℃
with monthly variation from -8.6 ℃ in January to 3.1 ℃ in July (Figure 2). The
average annual precipitation at the Yangbajain Station in the valley is about 427 mm.
The catchment has a summer (June-August) monsoon with 73% of the yearly



precipitation, while the rest of the year is dry with only 1% of the yearly precipitation
occurring in winter (December-February) (Figure 2).
The average annual streamflow is 277.7 mm, and the intra-annual distribution of
streamflow is uneven (Figure 2). In summer, streamflow is recharged mainly by
monsoon rainfall and meltwater, which accounts for approximately 63% of the yearly
streamflow (Figure 2). The streamflow in winter with only 4% of the yearly
streamflow (Figure 2) is only recharged by groundwater, which is greatly affected by
the freeze-thaw cycle of frozen ground and the active layer (Liu et al., 2011).
**2.2. Data**
Daily streamflow and precipitation data at the four hydrological Stations (Figure 1a)
during the period 1979-2013 are collected from the Tibet Autonomous Region
Hydrology and Water Resources Survey Bureau. The monthly meteorological data at
the three weather stations (Figure 1a) are obtained from the China Meteorological
Data Sharing Service System (http://data.cma.cn/) for the years from 1979 to 2013. In
this study, the method of meteorological data extrapolation by Prasch et al. (2013) is
adopted to obtain the discretisized air temperature (with cell size as 1 km×1 km) of
the Lhasa River Basin based on the air temperature of the three stations assuming a
linear lapse rate. The mean monthly lapse rate is set to 0.44 ℃/100m for elevations
below 4,965 m and 0.78 ℃/100m for elevations above 4,965 m in the catchment
(Wang et al., 2015).
The glaciers and frozen ground data are provided by the Cold and Arid Regions



Science Data Center (http://westdc.westgis.ac.cn/). The distribution, area and volume
of glaciers are based on the First and Second Chinese Glacier Inventory in 1960 and
2009 (Mi et al., 2002; Liu et al., 2014) (Figure 1b). The distribution and classification
of frozen ground (Figure 1c) are collected from the twice maps of frozen ground on
the TP (Li and Cheng, 1996; Zou et al., 2017).
The latest Level‑3 monthly mascon solutions (CSR, Save et al., 2016) was used to
detect terrestrial water storage (TWS, total vertically-integrated water storage)
changes for the period from January 2003 to December 2015 with spatial sampling of
$0.5°{\times}0.5°$ from the Gravity Recovery and Climate Experiment (GRACE) satellite.
The time series of 2003~2015 for snow water equivalent (SWE), total soil moisture
(SM, layer 0~200cm) from the dataset (GLDAS_Noah2.1, https://disc.gsfc.nasa.gov/)
were adopted for derivation of the groundwater storage (GWS) (Richey et al., 2015).
**2.3. Methods**
*2.3.1. Statistical methods for assessing streamflow changes*
The Mann-Kendall (MK) test, which is suitable for data with non-normally
distributed or nonlinear trends, is applied to detect trends of hydro-meteorological
time series (Mann, 1945; Kendall, 1975). To remove the serial correlation from the
examined time series, a Trend-Free Pre-Whitening (TFPW) procedure is needed prior
to applying the MK test (Yue et al., 2002). A more detailed description of the
Trend-Free Pre-Whitening (TFPW) approach was provided by Yue et al. (2002).
Gray relational analysis was aimed to find the major climatic or hydrological





factors that influenced an objective variable (Liu et al., 2005; Wang et al., 2013). In
this paper, gray relational analysis is used to investigate the main climatic factor
impacting the streamflow.
*2.3.2. Baseflow separation*

In this paper, the most widely used one-parameter digital filtering algorithm is

adopted for baseflow separation (Lyne and Hollick, 1979). The filter equation is
expressed as

$$q_t = \alpha q_{t-1} + \frac{1+\alpha}{2}(Q_t - Q_{t-1}) \tag{1}$$

$$b_t = Q_t - q_t \tag{2}$$

where $q_t$ and $q_{t-1}$ are the filtered quick flow at time step $t$ and $t$-1, respectively; $Q_t$ and
$Q_{t-1}$ are the total runoff at time step $t$ and $t$-1; $\alpha$ is the filter parameter, ranging from
0.9 to 0.95; $b_t$ is the filtered baseflow.
*2.3.3. Determination of active groundwater storage*

The method of recession flow analysis is widely used to investigate the baseflow

recession characteristics and the storage-discharge relationship of catchments (Lyon et
al., 2009; Lyon and Destouni, 2010; Sjöberg et al., 2013; Lin and Yeh., 2017; Gao et
al., 2017). Physical considerations based on hydraulic groundwater theory suggest
that the groundwater storage in a catchment can be approximated as a power function
of baseflow rate at the catchment outlet (Brutsaert, 2008)

$$S = Ky^m \tag{8}$$

where $S$ is the volume of active groundwater storage (abbreviated as groundwater



storage in the following context) in the catchment aquifers (see in Figure 3). The
active groundwater storage $S$ is defined as the storage that controls streamflow
dynamics assuming that streamflow during rainless periods is a function of catchment
storage (Kirchner, 2009; Staudinger, 2017); $K$, $m$ are constants depending on the
catchment physical characteristics, and $K$ is the baseflow recession coefficient,
represented the time scale of the catchment streamflow recession process; $y$ is the rate
of baseflow in the stream.

During dry season without precipitation and other input events, the flow in a stream

can be assumed to depend solely on the groundwater storage from the upstream
aquifers (Brutsaert, 2008; Lin and Yeh, 2017). For such baseflow conditions, the
conservation of mass equation can be represented as

$$\frac{dS}{dt} = -y \tag{9}$$

where $t$ is the time. Substitution of equation (8) in equation (9) yields (Brutsaert and
Nieber, 1977)

$$-\frac{dy}{dt} = ay^b \tag{10}$$

where $dy/dt$ is the temporal change of the baseflow rate during recessions, and the
constants $a$ and $b$ are called the recession intercept and recession slope of plots of
$-dy/dt$ versus $y$ in log-log space, respectively. The parameters of $K$ and $m$ in equation
(8) can be expressed by $a$ and $b$, where $K = 1/\left[a(2-b)\right]$ and $m = 2-b$ (Gao et al.,
2017). In the storage discharge relationship, the aquifer responds as a linear reservoir
if $b=1$, and as nonlinear reservoir if $b\neq 1$.



In our study, the baseflow recession data are selected from the streamflow
hydrographs, which remarkably decline for at least 3 days after rainfall ceases and
remove the first 2 days to avoid the impact of storm flow (Brutsaert and Lopez, 1998).
A variable time interval $\Delta t$ is used to properly scale the observed drop in streamflow
to avoid discretization errors on $-dy/dt \sim y$ plot due to measurement noise, especially in
the log-log space (Rupp and Selker, 2006; Kirchner, 2009). Then the constants $a$ and $b$
are fitted by using a nonlinear least squares regression through all data points of
$-dy/dt$ versus $y$ in log-log space for all years to avoid the difficulty of defining a lower
envelop of the scattered points (Lyon et al., 2009). Theoretically, one can fit a line of
slope $b$ to recession flow data graphed in this manner and determine aquifer
characteristics from the resulting value of $a$ (Rupp and Selker, 2006). That is to say,
with a fixed slope $b$ during recessions, it should be possible to observe the changes in
catchment aquifer properties by fitting the intercept $a$ as a variable across different
years. Since the values of $K$ and $m$ can be calculated by fitting recession intercept $a$
and the fixed slope $b$, the average groundwater storage $S$ for dry season can be
obtained through equation (8) based on average rate of baseflow.
**3. Results**
**3.1. Assessment of streamflow changes**
The annual streamflow of the Yangbajain Catchment shows an increasing trend at
the 5% significance level with a mean rate of about 12.30 mm/10a over the period
1979-2013 (Table 1 and Figure 4a). Meanwhile, annual mean air temperature exhibits

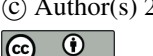



an increasing trend at the 1% significance level with a mean rate of about 0.28 ℃/10a
(Table 1 and Figure 5a). However, annual precipitation has a nonsignificant trend
during this period (Table 1 and Figure 5b).
As annual streamflow increases significantly, it is necessary to analyze to what
extent the changes in the two components (quick flow and baseflow) lead to
streamflow increases. Based on the baseflow separation method, the annual mean
baseflow contributes about 59% of the annual mean streamflow in the catchment. The
MK test shows that annual baseflow exhibits a significant increasing trend at the 1%
level with a mean rate of about 10.95 mm/10a over the period 1979-2013 (Table 1 and
Figure 4b). But the trend is statistically nonsignificant for annual quick flow in the
same period (Table 1). The increasing trends between the baseflow and streamflow
are very close, indicating that the increase in baseflow is the main contributor to
streamflow increases.
Furthermore, gray relational analysis is applied to the catchment to identify the
major climatic factors for the increasing streamflow. The result shows that the air
temperature has the higher gray relational grade at annual scale (Table 2). This
indicates that the air temperature acts as a primary factor for the increased streamflow
as well as the baseflow.
The annual streamflow and baseflow significantly increase due to the rising air
temperature over the period 1979-2013. However, there are diverse intra-annual
variation characteristics for streamflow as well as the two streamflow components





during the period. Streamflow in spring (March to May), autumn (September to
November) and winter (December to February) show increasing trends at least at the
5% significance level (Figure 6a, 6c and 6d), while streamflow in summer (June to
August) has a nonsignificant trend during this period (Figure 6b). Baseflow also
increases significantly in spring, autumn and winter (Figure 6a, 6c and 6d). The trend
is statistically nonsignificant for baseflow in summer (Figure 6b). Quick flow exhibits
nonsignificant trend for all seasons (Table 1). As to the meteorological factors, mean
air temperature in all seasons increase significantly at the 1% level especially during
winter with the rate of about 0.51 ℃/10a (Table 1 and Figure 7), whereas precipitation
in each season shows nonsignificant trend during these years (Table 1). The gray
relational analysis shows that the air temperature is the critical climatic factor for the
changes in streamflow and baseflow in all seasons (Table 2).

Compared with monsoon rainfall as the main water source for summer runoff, the

corresponding contribution of glacial meltwater to the streamflow only accounts for
max. 11% in the catchment (Prasch et al., 2013). Moreover, the summer meltwater
partly infiltrates into soils and will be stored in aquifers. This can explain why it is
statistically nonsignificant for summer runoff.
**3.2. Estimation of groundwater storage by baseflow recession analysis**

Using the data selection procedure mentioned in the section 2.3.3, we adopted daily

streamflow and precipitation records in autumn and early winter (September to
December) in which the hydrograph with little precipitation usually declines


consecutively and smoothly. The fitted slope *b* is equal to 1.79 through the nonlinear
least square fit of equation (10) for all data points of *-dy/dt* versus *y* in log-log space
during the period 1979-2013. Moreover, for each decade or year, the intercept *a* could
be fitted by the fixed slope *b*=1.79. Then, the values of *K* and *m* for each decade or
year can be determined. And the groundwater storage *S* for each year can be directly
estimated from the average rate of baseflow during a recession period through
equation (8).

Figure 8 shows the results of the nonlinear least square fit for each decade's

recession data from the 1980s, 1990s and 2000s, respectively. As shown in Figure 8,
the recession data points and fitted recession curves of each decade gradually move
downward as time goes on. This indicates that, with a fixed slope *b*, the intercept *a*
gradually decreases and recession coefficient *K* increases accordingly. The values of
recession coefficient *K* for each decade are 77 $\mathrm{mm}^{0.79}\mathrm{d}^{0.21}$, 84 $\mathrm{mm}^{0.79}\mathrm{d}^{0.21}$ and 103
$\mathrm{mm}^{0.79}\mathrm{d}^{0.21}$. Furthermore, Figure 9a shows the inter-annual variation of recession
coefficient *K* during the period 1979-2013. In total, though there are some large
fluctuations or even a rather large decrease at the beginning of the 1990s, the overall
increasing trend of 7.70 $(\mathrm{mm}^{0.79}\mathrm{d}^{0.21})$/10a at a significance level of 5% is similar to the
results obtained from decade analysis. This long-term variation of recession
coefficient *K* from September to December indicates that baseflow recession during
autumn and early winter gradually slows down in the catchment.

According to the results of decade data fit (see in Figure 8), the mean values of





groundwater storage $S$ estimated for each decade are 130 mm, 148 mm and 188 mm
for the 1980s, 1990s and 2000s. The trend analysis suggests that the groundwater
storage $S$ shows an increasing trend at the 5% significance level with a rate of about
19.32 mm/10a during the period 1979-2013 (Figure 9b). This indicates that
groundwater storage has been enlarged. The annual trend of groundwater storage $S$
from 1979 to 2013 is consistent with the values across decades. The inter-annual
variation of groundwater storage $S$ is also similar with recession coefficient $K$ (Figure
9a and 9b). The decreased trend of anomalies changes of groundwater storage (GWS)
estimated by the GRACE data is consistent with the annual trend of $S$ during
2003~2015 (Figure 9b). And the reduced volume of groundwater between GWS and $S$
are also similar (~100-120 mm).
**4. Discussions**

The results have revealed that the increase of streamflow especially in dry season is

tightly related with climate warming. It is obviously that both glacier retreat and
frozen ground degradation in a warmer climate can significantly alter the mechanism
of streamflow. In the Yangbajain Catchment as well as the whole Lhasa River Basin,
it is experiencing a noticeable glacier retreat and frozen ground degradation during the
past decades (Table 3). For instance, according to the twice map of frozen ground
distribution on the TP (Li and Cheng, 1996; Zou et al., 2017), the areal extent of
permafrost in the Yangbajain catchment has decreased by 406 km$^2$ (15.3%) over the
past 22 years; the corresponding areal extent of seasonal frozen ground has increased



by 406 km$^2$ (15.3%) with the degradation of permafrost.
According to the new map of permafrost distribution on the Tibetan Plateau (Zou et
al., 2017), the coverages of permafrost and seasonally frozen ground in each
sub-catchment (especially the Lhasa sub-catchments) are comparable to that in the
Yangbajain Catchment; but the coverage of glaciers in the three catchments is far
lower than that in the Yangbajain Catchment according to the First Chinese Glacier
Inventory (Mi et al., 2002) (Table 3). The MK test showed that, in all the four
catchments, the annual mean air temperature had significant increases at the 1%
significance level (Figure 4) while the annual precipitation showed nonsignificant
trends (Table 4). The annual streamflow of the three Lhasa, Pangdo and Tangga
Catchments all had nonsignificant trends, while the annual streamflow of the
Yangbajain Catchment showed an increasing trend at the 5% significance level with a
mean rate of about 12.30 mm/10a during the period. Ye et al. (1999) stated that when
glacier coverage is greater than 5%, glacier contribution to streamflow starts to show
up. This indicates that, in the Yangbajain Catchment, the increased streamflow is
mainly fed by the accelerated glacier retreat rather than frozen ground degradation.
This conclusion is also consistent with previous results by Prasch et al. (2013), who
suggested that the contribution of accelerated glacial meltwater to streamflow would
bring a significant increase in streamflow in the Yangbajain Catchment. Thus it is
reasonable to attribute annual streamflow increases to the accelerated glacier retreat as
the consequence of increasing annual air temperature.





Although permafrost degradation is not the controlling factor for the increase of
streamflow, a rational hypothesis is that increased groundwater storage $S$ in autumn
and early winter is associated with frozen ground degradation, which can enlarge
groundwater storage capacity (Niu et al., 2016). Figure 3 depicts the changes of
surface flow and groundwater flow paths in a glacier fed catchment, which is
underlain by frozen ground under past climate and warmer climate, respectively. As
frozen ground extent continues to decline and active layer thickness continues to
increase in the valley, the enlargement of groundwater storage capacity can provide
enough storage space to accommodate the increasing meltwater that may percolate
into deeper aquifers (Figure 3). Then, the increase of groundwater storage in autumn
and early winter allows more groundwater discharge into streams as baseflow, and
lengthens the recession time as indicated by recession coefficient $K$. This leads to the
increased baseflow and slow baseflow recession in autumn and early winter, as is
shown in Figure 6c, 6d and Figure 9a. In the late winter and spring, the increase of
baseflow (Figure 6d and 6a) can be explained by the delayed release of increased
groundwater storage.
Thus, as the results of climate warming, river regime in this catchment has been
altered significantly. On the one hand, permafrost degradation is changing the aquifer
structure that controls the storage-discharge mechanism, e.g., catchment groundwater
storage increases at about 19.32 mm/10a. On the other hand, huge amount of water
from glacier retreat is contributing to the increase of streamflow and groundwater





storage. For example, the annual streamflow of the Yangbajain Catchment increases
with a mean rate of about 12.30 mm/10a during the past 50 years. However, the total
glacial area and volume have decreased by 38.05 km$^2$ (12.0%) and 4.73×10$^9$ m$^3$
(26.2%) over the period 1960-2009 (Figure 10) according to the Chinese Glacier
Inventories. Hence, the reduction rate of glacial volume is 9.46×10$^7$ m$^3$/a (about 357.7
mm/10a) on average during the past 50 years. In the ablation on continental type
glaciers in China, evaporation (sublimation) always takes an important role, however,
annul amount of evaporation is usually less than 30% of the total ablation of glaciers
in the high mountains of China (Zhang et al., 1996). Given the 30% reduction in
glacial melt, there is still a large water imbalance between melt-derived runoff and the
actually increase of runoff and groundwater storage.
So the considerable water imbalance (estimated at least to be 5.79×10$^7$ m$^3$/a)
provides a perspective about the deep subsurface leakage through the fault zone in the
Yangbajain Catchment. Our results imply that more than 60% of glacial meltwater
would be lost by subsurface leakage. In fact, the north-south trending fault in the
Yangbajain Catchment plays a significant role on accessing groundwater flow through
deep pathway (Jiang et al., 2016).
**5. Conclusions**
In this study, the changes of hydro-meteorological variables were evaluated to
identify the main climatic factor for streamflow increases in the Yangbajain
Catchment, a sub-basin with the largest glacier coverage and a widespread frozen



ground in the Lhasa River Basin in the south-central TP. We analyzed the changes of
streamflow components through baseflow separation method. We quantified baseflow
recession and active groundwater storage in autumn and early winter by recession
flow analysis, and discussed the seasonal variations of baseflow in response to the
changes in active groundwater storage.
We find that the annual streamflow especially the annual baseflow increases
significantly, and the rising air temperature acts as a primary factor for the increased
runoff. The increased streamflow is mainly fed by the accelerated glacier retreat due
to climate warming. The decreased glacial volume has supplied large quantities of
glacial meltwater which recharge aquifers and reside in temporary storage during
summer, and then release as baseflow during the following seasons. Moreover, frozen
ground degradation would enlarge groundwater storage capacity, and then provide
more storage spaces for the meltwater. This can explain why baseflow volume
increases and baseflow recession slows down in autumn and early winter. At last we
find that there is a large water imbalance ($> 5.79 \times 10^7$ m$^3$/a) between melt-derived
runoff and the actually increase of runoff and groundwater storage, which suggests
more than 60% of the reduction in glacial melt should be lost by subsurface leakage
through the fault zone in the Yangbajain catchment.
This study provides a fundamental understanding of the changes in streamflow and
groundwater storage under a warming climate. It is of great importance to predict the
effects of future climate changes on water resources and hydrological processes in



highly glacier-fed and large-scale frozen ground regions. More methods (e.g.,
hydrological isotopes) should be adopted to quantify the contribution of glaciers
meltwater and permafrost degradation to streamflow, and to explore the change of
groundwater storage capacity as frozen ground continues to degrade.
**Acknowledgements:**
This work was supported by the National Natural Science Foundation of China
(NSFC) (grants 91647108, 91747203), the Science and Technology Program of Tibet
Autonomous Region (2015XZ01432), and the Special Fund of the State Key
Laboratory of Hydrology-Water Resources and Hydraulic Engineering (no

20185044312).

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

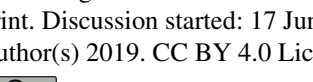

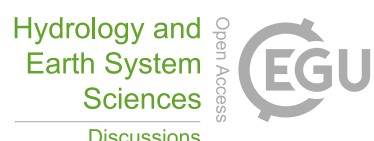

**Table 1.** Mann-Kendall trend test with trend-free pre-whitening of seasonal and annual mean air temperature (℃), precipitation (mm), streamflow (mm), baseflow (mm) and quick flow (mm) from 1979 to 2013.

| | Air temperature | | Precipitation | | Streamflow | | Baseflow | | Quick flow | |
|---|---|---|---|---|---|---|---|---|---|---|
| | $Z_C$ | $\beta$ (℃/a) | $Z_C$ | $\beta$ (mm/a) | $Z_C$ | $\beta$ (mm/a) | $Z_C$ | $\beta$ (mm/a) | $Z_C$ | $\beta$ (mm/a) |
| Spring | 2.73** | 0.026 | 0.90 | 0.290 | 3.05** | 0.206 | 2.99** | 0.147 | 0.98 | 0.042 |
| Summer | 2.63** | 0.013 | 1.30 | 2.139 | 0.92 | 0.549 | 1.27 | 0.429 | 0.50 | 0.128 |
| Autumn | 2.65** | 0.024 | -0.68 | -0.395 | 2.46* | 0.546 | 2.96** | 0.476 | 0.80 | 0.074 |
| Winter | 3.49** | 0.051 | -0.46 | -0.014 | 3.08** | 0.204 | 2.13* | 0.145 | 1.39 | 0.016 |
| Annual | 4.48** | 0.028 | 1.28 | 2.541 | 2.07* | 1.230 | 2.70** | 1.095 | 0.77 | 0.327 |

Comment: the symbols of $Z_C$ and $\beta$ mean the standardized test statistic and the trend magnitude, respectively; positive values of $Z_C$ and $\beta$ indicate the upward trend, whereas negative values indicate the downward trend in the tested time series; the symbols of asterisks *and ** mean statistically significant at the levels of 5% and 1%, respectively.

**Table 2.** Gray relational grades between the streamflow/baseflow and climate factors (precipitation and air temperature) in the Yangbajain Catchment at both annual and seasonal scales. Bold text shows the higher gray relational grade in each season.

| | $G_{oi}$ with the streamflow | | $G_{oi}$ with the baseflow | |
|---|---|---|---|---|
| | Precipitation | Air temperature | Precipitation | Air temperature |
| Spring | 0.690 | **0.778** | 0.713 | **0.789** |
| Summer | 0.689 | **0.784** | 0.680 | **0.776** |
| Autumn | 0.653 | **0.667** | 0.648 | **0.680** |
| Winter | 0.742 | **0.886** | 0.748 | **0.895** |
| Annual | 0.675 | **0.727** | 0.665 | **0.729** |

Comment: $G_{oi}$ is the gray relational grade between the streamflow/baseflow and climate factors. The importance of each influence factor can be determined by the order of the gray relational grade values. The influence factor with the largest $G_{oi}$ is regarded as the main stress factor for the objective variable.








Table 3. The coverage of glaciers and frozen ground in four catchments of the Lhasa River Basin

| Stations | Area (km²) | Glaciers(1960) Area (km²) | Coverage (%) | Glaciers(2009) Area (km²) | Coverage (%) | Permafrost (1996) Area (km²) | Coverage (%) | Permafrost (2017) Area (km²) | Coverage (%) | Seasonally frozen ground (1996) Area (km²) | Coverage (%) | Seasonally frozen ground (2017) Area (km²) | Coverage (%) |
|---|---|---|---|---|---|---|---|---|---|---|---|---|---|
| Lhasa | 26233 | 349.26 | 1.3 | 347.14 | 1.3 | 10535 | 40.2 | 9783 | 37.3 | 15698 | 59.8 | 16450 | 62.7 |
| Pangdo | 16425 | 345.24 | 2.1 | 339.90 | 2.1 | 8666 | 52.7 | 8242 | 50.2 | 7762 | 47.3 | 8184 | 49.8 |
| Tangga | 20152 | 348.12 | 1.7 | 342.27 | 1.7 | 10081 | 50.0 | 9432 | 46.8 | 10071 | 50.0 | 10720 | 53.2 |
| Yangbajain | 2645 | 316.31 | 12.0 | 278.26 | 10.5 | 1352 | 51.1 | 946 | 35.8 | 1293 | 48.9 | 1699 | 64.2 |

Table 4. Mann-Kendall trend test with trend-free pre-whitening of annual mean air temperature (°C), precipitation (mm) and streamflow (mm) in four catchments of the Lhasa River Basin

| | Air temperature $Z_C$ | $\beta$ (°C/a) | Precipitation $Z_C$ | $\beta$ (mm/a) | Streamflow $Z_C$ | $\beta$ (mm/a) |
|---|---|---|---|---|---|---|
| Lhasa | 6.07** | 0.028 | 1.16 | 1.581 | 1.09 | 1.420 |
| Pangdo | 6.19** | 0.026 | 0.89 | 1.435 | 0.30 | 0.223 |
| Tangga | 7.35** | 0.021 | 1.48 | 2.005 | -0.62 | -0.531 |
| Yangbajain | 4.48** | 0.028 | 1.28 | 2.541 | 2.07* | 1.230 |





**Figure captions**
**Figure 1.** (a) The location, (b) elevation distribution, and (c) glacier and frozen
ground distribution (Zou et al., 2017) in the Yangbajain Catchment of the Lhasa River
Basin in the TP.
**Figure 2.** Seasonal variation of streamflow ($R$), mean air temperature ($T$), and
precipitation ($P$) in the Yangbajain Catchment.
**Figure 3.** Diagram depicting surface flow and groundwater flow due to glacier melt
and frozen ground thaw under (a) past climate and (b) warmer climate. Blue lines
with arrows are conceptual surface flow paths. Red lines with arrows are conceptual
groundwater flow paths (after Evans and Ge. (2017)).
**Figure 4.** Variations of annual (a) streamflow and (b) baseflow from 1979 to 2013.
**Figure 5.** Variations of annual (a) mean air temperature and (b) precipitation from
1979 to 2013.
**Figure 6.** Variations of seasonal streamflow and baseflow in (a) spring, (b) summer,
(c) autumn, and (d) winter from 1979 to 2013.
**Figure 7.** Variations of seasonal mean air temperature in (a) spring, (b) summer, (c)
autumn, and (d) winter from 1979 to 2013.
**Figure 8.** Recession data points of $-dy/dt$ versus $y$ and fitted recession curves by
decades in log-log space. The black point line, dotted line, and solid line represent
recession curves in the 1980s, 1990s, and 2000s, respectively.
**Figure 9.** Variations of (a) the recession coefficient $K$ and (b) groundwater storage $S$





from 1979 to 2013.
**Figure 10.** The total area and volume of glaciers in the Yangbajain Catchment in 1960
and 2009.

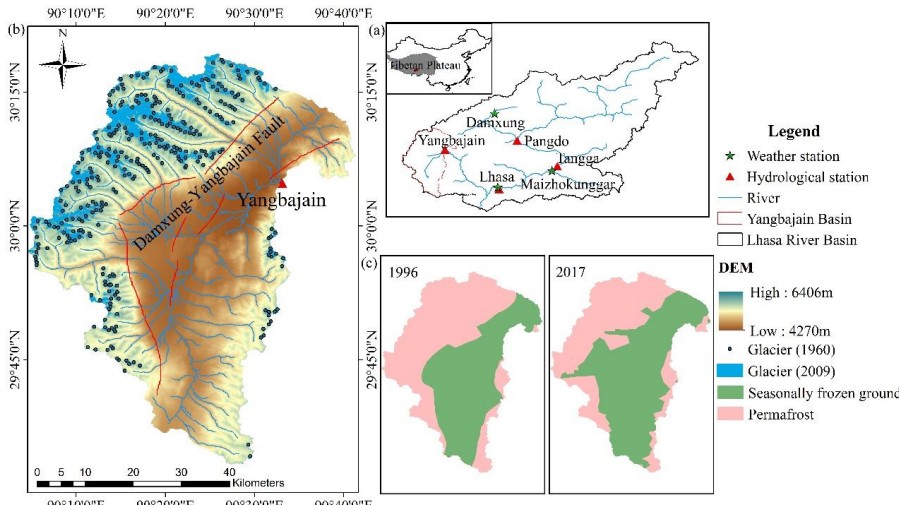


**Figure 1.** (a) The location, (b) elevation and glacier distribution for the twice Chinese
Glacier Inventory, only the location of glacier snouts in 1960 were provided in the
first Chinese Glacier Inventory, and the boundaries of glaciers were shown in the
second Chinese Glacier Inventory, and (c) twice maps of frozen ground distribution
(Li and Cheng, 1996; Zou et al., 2017) in the Yangbajain Catchment.

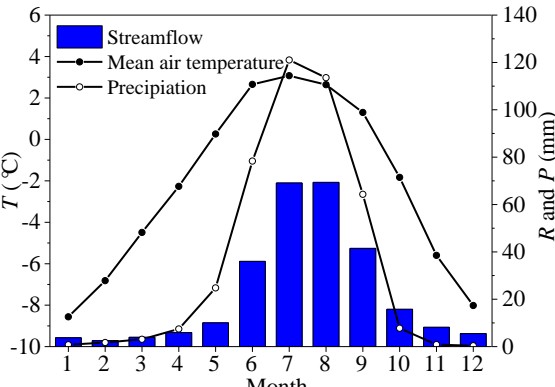


**Figure 2.** Seasonal variation of streamflow ($R$), mean air temperature ($T$), and

precipitation ($P$) in the Yangbajain Catchment.


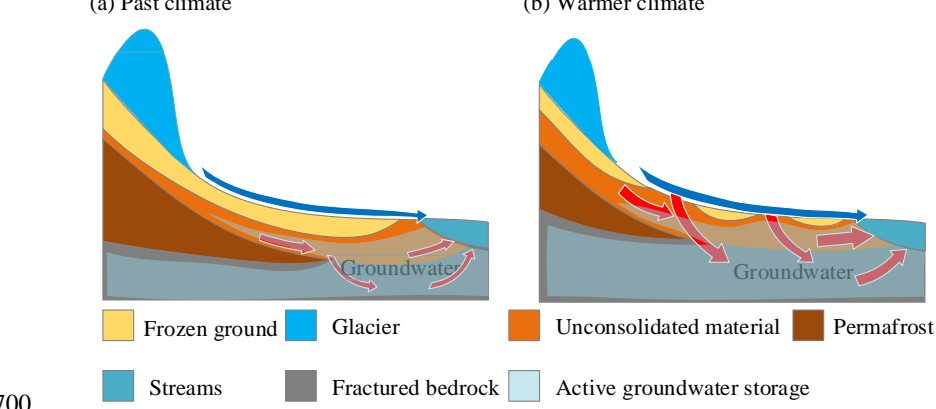


**Figure 3.** Diagram depicting surface flow and groundwater flow due to glacier melt

and frozen ground thaw under (a) past climate and (b) warmer climate. Blue lines

with arrows are conceptual surface flow paths. Red lines with arrows are conceptual

groundwater flow paths (after Evans and Ge. (2017)).





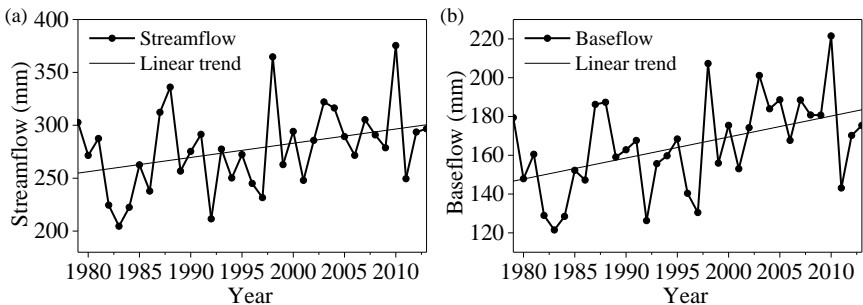


**Figure 4.** Variations of annual (a) streamflow and (b) baseflow from 1979 to 2013.


**Figure 5.** Variations of annual (a) mean air temperature and (b) precipitation from
1979 to 2013.



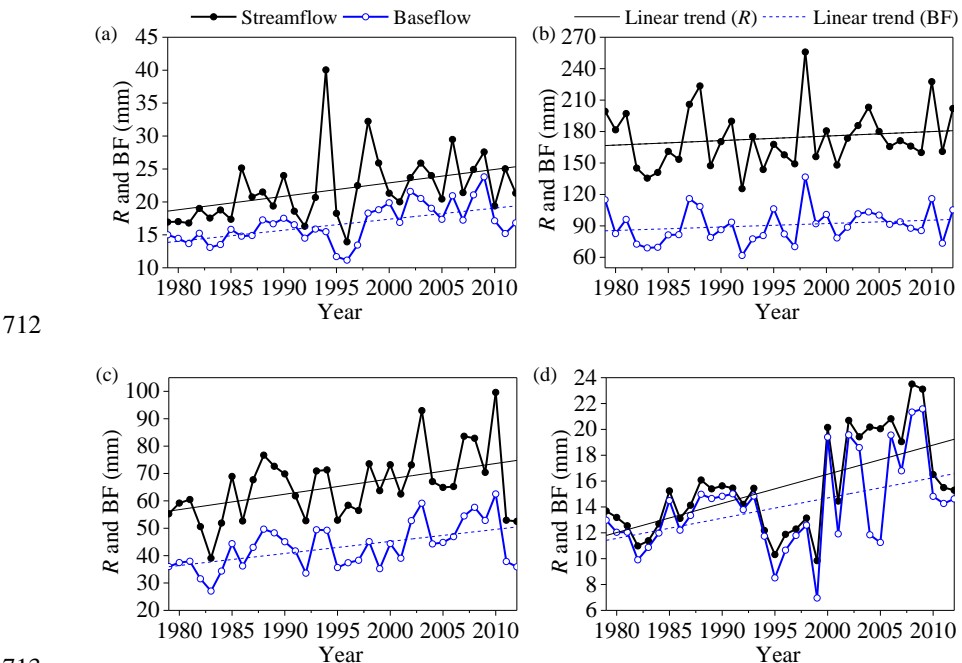

**Figure 6.** Variations of seasonal streamflow and baseflow in (a) spring, (b) summer,

(c) autumn, and (d) winter from 1979 to 2013.






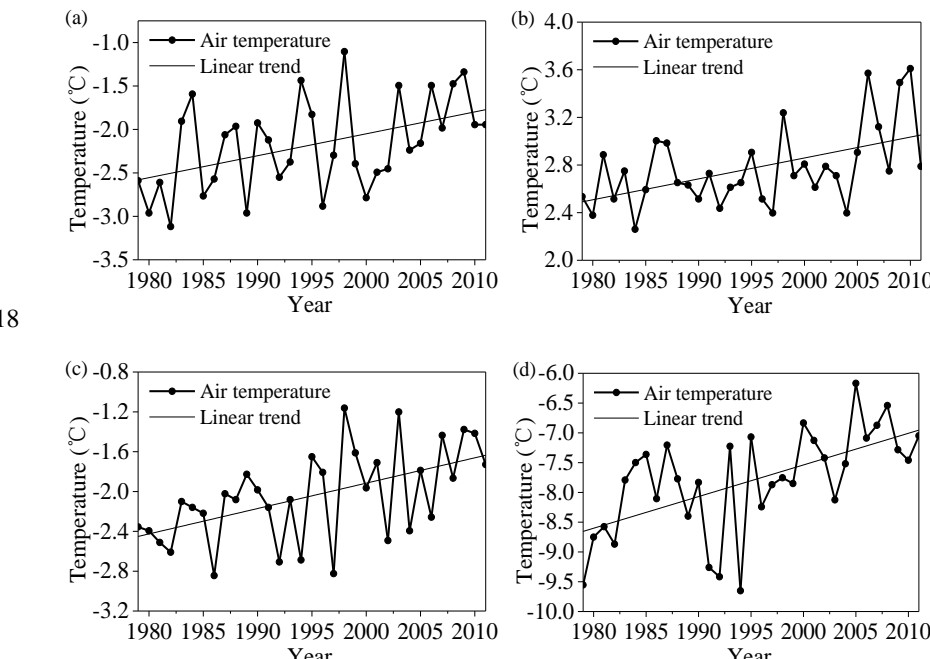


**Figure 7.** Variations of seasonal mean air temperature in (a) spring, (b) summer, (c)

autumn, and (d) winter from 1979 to 2013.



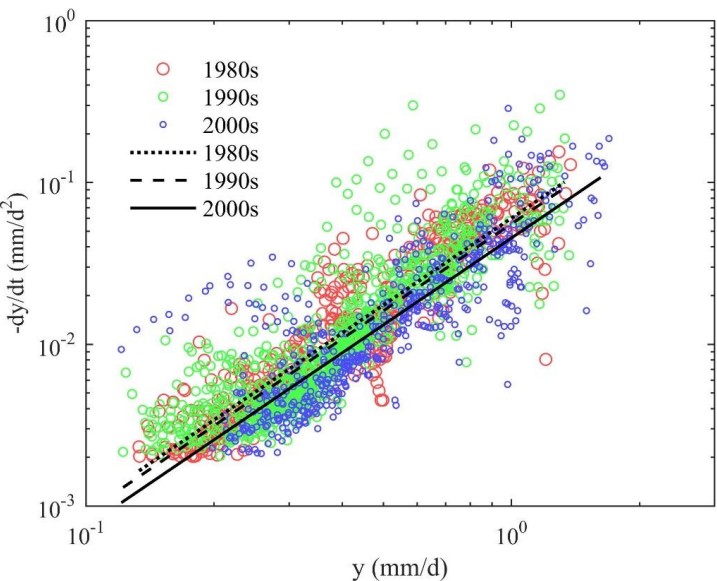


**Figure 8.** Recession data points of *-dy/dt* versus *y* and fitted recession curves by

decades in log-log space. The black point line, dotted line, and solid line represent

recession curves in the 1980s, 1990s, and 2000s, respectively.


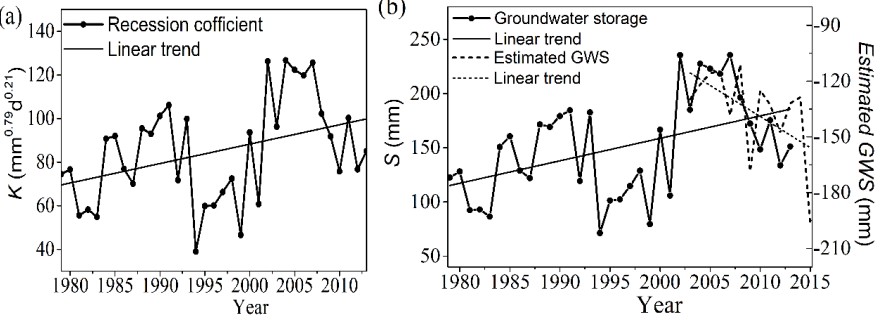


**Figure 9.** Variations of (a) the recession coefficient *K* and (b) the estimated

groundwater storage *S* from 1979 to 2013 and the estimated groundwater storage

change from 2003 to 2015 by GRACE data.






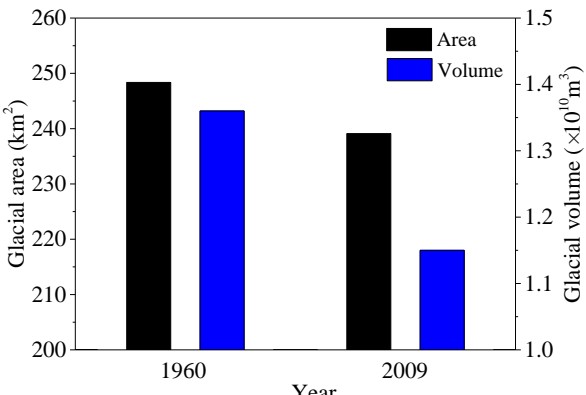

**Figure 10.** The total area and volume of glaciers in the Yangbajain Catchment in

1960 and 2009.
