# Peer review of "Understanding the effects of climate warming on streamflow and active"

_Hydrology and Earth System Sciences, 2019_

## Referee Comment (RC1) · Anonymous Referee #1 · 6 Jul 2019

This paper presents the temperature, precipitation and stream variation in the Yangbajain catchment. Interestingly, the estimate the base flow and connect the baseflow variation with the climate change. This is important for the local water resources management and well as for the global groundwater-climate change research. But it should be accepted after a minor revision. My major comments are: 1. The accuracy of base flow and groundwater storage estimation. As I pointed out in the specific comment, the authors should provide more evidences to show the estimated groundwater storage are correct. 2. The explanation on the glacier loss should be deleted. Please see the specific comment (Line 408-413). 3. The schematic model (Figure 3). (1) The glacier thickness should increase with the altitude; (2) 'Unconsolidated material'

changes 'Unconsolidated soil layer'; (3) Take care of the width of the arrows. Specific comments: Line 115&117 What is the method difference between Lyon et al. (2009) and Kirchner et al. (2009)? And what is the latest advance of the recession analysis? Please clarify. Line 163, 164&168. Please describe the number clearly on the period as well as the hydrologic station. Line 169-171 How do you get the number of 63% from Fig. 2. And I do not think you can get this number easily only with the data of temperature, precipitation amount and runoff. Line 286-288 The higher grade relational grade is found at the annual scale, how can you say the air temperature also acts a primary role for the base flow? Line 339-344 I suggest to shift these sentences above the lines 335-339. Before discussing the trend of the groundwater storage, you should firstly explain the obtained results of groundwater storage are reasonable. I also ask the authors to give more explanation on their obtained groundwater storage, because it does seems consistent between the Grace data and your data. Could the authors give more evidences of the monitored groundwater level? Line 356-370 I understand the authors try to draw the conclusion 'the increased streamflow is mainly fed by the accelerated glacier retreat rather than frozen ground degradation' through the comparison between four catchments. This is something kind of 'circumstantial evidence'. Could you explain why the frozen ground degradation does not increase the streamflow? Line 408-413. This is quite arbitrary. Although the estimation of glacier loss is reasonable, the loss can be explained in many ways. For example, it could be delivered through the different pathways of shallow aquifer; and it could be exchanged with the aquifers outside the studied region. Sure, it may also infiltrate into the deep fault. But all of these hypotheses need evidences. If you take the one of deep circulation, you should describe clearly the hydrogeologic features of the fault. Is it conductive or not? What is the depth of it? What is the groundwater flow direction inside it? Could you provide the hydrogeologic section map here? If the authors could not provide the discussion above, I suggest the authors to delete this paragraph and leave the glacier loss as an open discussion question here.

---

## Referee Comment (RC2) · Anonymous Referee #2 · 10 Jul 2019

Journal: HESS Title: Quantifying streamflow and active groundwater storage in response to climate warming in an alpine catchment, upper Lhasa River MS No.:HESS_2019_302

In this work, Lin et al. has investigated the changes in streamflow regimes and climate factors are evaluated based on hydro-meteorological observations from 1979 to 2013. The work is very interesting. This study provides a perspective to clarify the impact of glacial retreat and frozen ground degradation on hydrological processes, which fundamentally affects the water supply and the mechanisms of streamflow generation and change. However, I have some issues with this paper, which prevents me from giving

a positive recommendation.

1.The title of this paper is: Quantifying streamflow and active groundwater storage in response to climate warming in an alpine catchment, upper Lhasa River. However, the main content of this paper is the relationship between streamflow and active groundwater storage and temperature and precipitation. Moreover, the response of runoff on climate warming is not clearly quantified in this paper. So this topic may not be suitable for this article. 2.In this paper, the mechanism of hydrological process, hydrological cycle and the relationship between recharge and drainage of water in alpine region are not described in detail. Please add it. 3. " the annual streamflow especially the annual baseflow increases significantly, and the rising air temperature acts as a primary factor for the increased runoff. " . Climate warming has been a fact. Glacier could be reduced by the increasing of temperature is a fact, too. However, this conclusion should be for the ablation period only in your study areaïijĹCold regionsïijĽ. I suggest authors make a more detailed analysis of the Year, Month, the ablation period and freezing period, which may be more reasonable and interesting. 4.Diagram depicting surface flow and groundwater flow due to glacier melt and frozen ground thaw of Figure 3 should not be in the alpine region, at least not in the Qinghai-Tibet Plateau. I suggest that the author make major revisions according to the current studies. 5.This work has been found that the increased streamflow is mainly fed by the accelerated glacier retreat due to climate warming. There are many factors for the increase of streamflow. The accelerated glacier is just one of all factors. For example, the increase of precipitation, the degradation of frozen soil, the melting of underground ice, and the supply of supra-permafrost water. So I suggest that authors first figure out what is the main sources of streamflow in the study area? Then analyzed the contribution of the recharge sources to runoff based on the variation of all factors under the climate warming. Finally, the main reason for the increase for runoff is obtained. 6.This study also found that the decreased glacial volume has supplied large quantities of glacial meltwater which recharge aquifers and reside in temporary storage during summer, and then release as baseflow during the following seasons. So I suggest that the authors learn more about the mechanism of the hydrological process in the cold regions. 7.I don't think the discussion section is well written, so I think the discussion section may need to be re-written. 8.On the whole, the idea of this paper is very good, the conclusion of this paper is interesting, but the data support and supporting materials are lacking. In addition, the mechanism of water transformation in alpine region needs to be further studied.

---

## Author Comment (AC1) · 8 Sep 2019

Journal: HESS Title: Quantifying streamflow and active groundwater storage in response to climate warming in an alpine catchment, upper Lhasa River MS No.: HESS_2019_302 In this work, Lin et al. has investigated the changes in streamflow regimes and climate factors are evaluated based on hydro-meteorological observations from 1979 to 2013. The work is very interesting. This study provides a perspective

to clarify the impact of glacial retreat and frozen ground degradation on hydrological processes, which fundamentally affects the water supply and the mechanisms of streamflow generation and change. However, I have some issues with this paper, which prevents me from giving a positive recommendation. Response: Thank you for your valuable comments. Major revisions have been made to response to the reviewer's critiques. In the following, we provide point-by-point response to each reviewer comment (blue texts are our responses, while black texts are original comments).

1.The title of this paper is: Quantifying streamflow and active groundwater storage in response to climate warming in an alpine catchment, upper Lhasa River. However, the main content of this paper is the relationship between streamflow and active groundwater storage and temperature and precipitation. Moreover, the response of runoff on climate warming is not clearly quantified in this paper. So this topic may not be suitable for this article. Response: Yes, the title is somewhat unsuitable. There are not more evidences for quantifying the pathway and assessing the accurate contribution of each factor to runoff increasing. It tends to be a qualitative assessment of the effects of climate warming on hydrological processes. Thus in the revised version of the manuscript, we have changed the title as "Understanding the effects of climate warming on streamflow and active groundwater storage in an alpine catchment, upper Lhasa River".

2.In this paper, the mechanism of hydrological process, hydrological cycle and the relationship between recharge and drainage of water in alpine region are not described in detail. Please add it. Response: Yes, we have re-reviewed several latest or key studies in alpine regions. For example, Rogger et al. (2017)'s study about mountain permafrost, Xu et al. (2019)'s study about climate change on water budget in cryospheric-dominated watershed, Walvoord et al. (2007)'s analysis of increased groundwater to discharge by permafrost thawing in an arctic basin, and Su et al. (2016) and so on. Anyway, through reviewing these important studies, it helps us to re-organize the structure of our manuscript. We have re-written the section of introduction and conclusion and parts of other sections. In the manuscript, we have made it clear that in alpine regions, climate warming by triggering glacier retreat and permafrost thawing is changing hydrological processes of storage and discharge. However, direct measurement of the changing of permafrost depth or catchment aquifer storage is still difficult to perform at catchment scale. So quantitatively characterizing storage properties and sensitivity to climate warming in cold alpine catchments is desired. Hence, in this study, recession flow analysis is adopted to quantify active groundwater storage volume.

3. " the annual streamflow especially the annual baseflow increases significantly, and the rising air temperature acts as a primary factor for the increased runoff. ". Climate warming has been a fact. Glacier could be reduced by the increasing of temperature is a fact, too. However, this conclusion should be for the ablation period only in your study areaïij′LCold regionsïijĽ. I suggest authors make a more detailed analysis of the Year, Month, the ablation period and freezing period, which may be more reasonable and interesting. Response: Sure, it is important to analyze hydroclimatic responses in different seasons. In fact, we have added such contents in our manuscript. We found that there are diverse intra-annual variation characteristics for streamflow during the period. Streamflow in spring (March to May), autumn (September to November) and winter (December to February) show increasing trends at least at the 5% significance level (Figure 6a, 6c and 6d), while streamflow in summer (June to August) has a nonsignificant trend during this period (Figure 6b). Baseflow also increases significantly in spring, autumn and winter (Figure 6a, 6c and 6d). The trend is statistically nonsignificant for baseflow in summer (Figure 6b). As to the meteorological factors, mean air temperature in all seasons increase significantly at the 1% level especially during winter with the rate of about 0.51°C/10a (Table 1 and Figure 7), whereas precipitation in each season shows nonsignificant trend during these years (Table 1).

4.Diagram depicting surface flow and groundwater flow due to glacier melt and frozen ground thaw of Figure 3 should not be in the alpine region, at least not in the Qinghai-Tibet Plateau. I suggest that the author make major revisions according to the current studies. Response: Thank you very much for your suggestions. According to your suggestions and those of the other reviewer, we have made corresponding modifications according to the topography and distribution of glaciers and permafrost in the Yangbajain catchment. The details are available in 2.1 Study area of the manuscript. In Figure 1 depicting, we referred to the book of Ding et al. (2017) that is an introduction to hydrology in the cold regions especially in China.

5. This work has been found that the increased streamflow is mainly fed by the accelerated glacier retreat due to climate warming. There are many factors for the increase of streamflow. The accelerated glacier is just one of all factors. For example, the increase of precipitation, the degradation of frozen soil, the melting of underground ice, and the supply of supra-permafrost water. So I suggest that authors first figure out what is the main sources of streamflow in the study area? Then analyzed the contribution of the recharge sources to runoff based on the variation of all factors under the climate warming. Finally, the main reason for the increase for runoff is obtained. Response: thank you for your kindly suggestions. Many parts of this manuscript have been re-organized based on the reviewer's suggestions. The main water source for summer runoff in the study area is monsoon rainfall. And the runoff volume in summer account for 63% of the annual streamflow volume. As estimated by Prasch et al. (2013), the corresponding contribution of glacial meltwater to the streamflow only accounts for max. 11% in the catchment. Thus if precipitation increases/decreases significantly, runoff will change accordingly. However, in the catchment, precipitation in each season shows nonsignificant trend during these years (Table 1). The results of gray relational analysis indicate that the air temperature acts as a primary factor for the increased streamflow. As a results of climate warming, the areal extent of permafrost in the Yangbajain catchment has decreased by 406 km2 (15.3%) over the past 22 years, the total glacial area and volume have decreased by 38.05 km2 (12.0%) and $4.73 \times 109$ m3 (26.2%) over the period 1960-2009. All these changes have contributed to the changes of streamflow. At last, through parallel comparison of different sub-basins (Table 3 in the manuscript), we can indirectly conclude that the contribution of glacier retreat is much larger than frozen ground degradation. While the mostly significant effects of frozen ground degradation on runoff is that it can increase groundwater storage space and change the behavior of storage-discharge in the catchment. Similar results can be found in many other studies, e.g., Xu et al. (2019), Khadka et al. (2018) and Walvoord and Striegl (2007). For example, Walvoord and Striegl (2007) found that permafrost thawing in an arctic basin has resulted in a general upwards trend in groundwater contribution to streamflow of 0.7-0.9%/yr, however, with no pervasive change in total annual runoff. 6.This study also found that the decreased glacial volume has supplied large quantities of glacial meltwater which recharge aquifers and reside in temporary storage during summer, and then release as baseflow during the following seasons. So I suggest that the authors learn more about the mechanism of the hydrological process in the cold regions. Response: Yes, we have re-reviewed many references and added some of them in the revised version. Many parts of the manuscript have been re-written. See details in Response 2.

7.I don't think the discussion section is well written, so I think the discussion section may need to be re-written. Response: Thank you for your suggestion. The discussion section has been re-written as below. In this study, the changes of hydrometeorological variables were evaluated to identify the main climatic factor for streamflow increases in the cryospheric Yangbajain Catchment. We find that the annual streamflow especially the annual baseflow increases significantly, and the rising air temperature acts as a primary factor for the increased runoff. Furthermore, through parallel comparisons of sub-basins in the Lhasa River Basin, we indirectly presumed that the increased streamflow in the Yangbajain catchment is mainly fed by glacier retreat. Due to the climate warming, the total glacial area and volume have decreased by 38.05 km2 (12.0%) and 4.73×109 m3 (26.2%) in 1960-2009, and the areal extent of permafrost has degraded by 406 km2 (15.3%) in the past 22 years. As a results of permafrost degradation, groundwater storage capacity has been enlarged, which triggers a continuous increase of groundwater storage at a rate of about 19.32 mm/10a. This can explain why baseflow volume increases and baseflow recession slows down in autumn and early winter. At last we find that there is a large water imbalance (> 5.79×107 m3/a) between melt-derived runoff and the actually increase of runoff and groundwater storage, which suggests more than 60% of the reduction in glacial melt should be lost by subsurface leakage. However, the pathway of these leakage is still an open question for further studies. More methods (e.g., hydrological isotopes) should be adopted to quantify the contribution of glaciers meltwater and permafrost degradation to streamflow, and to explore the change of groundwater storage capacity as frozen ground continues to degrade.

8.On the whole, the idea of this paper is very good, the conclusion of this paper is interesting, but the data support and supporting materials are lacking. In addition, the mechanism of water transformation in alpine region needs to be further studied.' Response: Thank you for your positive comments. According to your suggestions, we have revised our topic as "Understanding the effects of climate warming on streamflow and active groundwater storage in an alpine catchment, upper Lhasa River". As we don't have more evidences for quantifying the pathway and assessing the accurately contribution of each factor to runoff increasing, we tend to present the manuscript as a qualitative assessment of the effects of climate warming on hydrological processes instead of a quantitative study. Moreover, we deleted some arbitrary conclusions. For instance, in the original version of the manuscript (Line 408-413.), we argue huge amount glacier loss is through deep fault. However, it is only a hypotheses and it still need further evidences. So we deleted this paragraph and leave the glacier loss as an open discussion question here. Anyway, we have made a major revision to present correct results and appropriate conclusions. Thank you again for you critiques.

References: Ding, Y. J., Zhang. S.Q., and Chen, R. S.: Introduction to hydrology in cold regions, Science Press, Beijing, China, 2017. (In Chinese). Khadka, N., Zhang, G., and Thakuri, S.: Glacial Lakes in the Nepal Himalaya: Inventory and Decadal Dynamics (1977–2017). Remote Sensing, 10, 1913, doi:10.3390/rs10121913, 2018. Prasch, M., Mauser, W., and Weber, M.: Quantifying present and future glacier melt-water contribution to runoff in a central Himalayan river basin, Cryosphere, 7(3), 889-904, doi:10.5194/tc-7-889-2013, 2013. Rogger, M., Chirico, G. B., Hausmann, H. Krainer, K. Brückl, E. Stadler, P. and Blöschl, G.: Impact of mountain permafrost on flow path and runoff response in a high alpine catchment, Water Resources Research, 53, 1288-1308, doi:10.1002/ 2016WR019341, 2017. Su, F., Zhang, L., Ou, T., Chen, D., Yao, T., Tong, K., and Qi, Y.: Hydrological response to future climate changes for the major upstream river basins in the Tibetan Plateau. Global and Planetary Change, 136, 82-95, doi:10.1016/j.gloplacha. 2015.10.012, 2016. Walvoord, M. A., and Striegl, R. G.: Increased groundwater to stream discharge from permafrost thawing in the Yukon River basin: Potential impacts on lateral export of carbon and nitrogen, Geophysical Research Letters, 34(12), 123-134, doi:10.1029/2007GL030216, 2007. Xu, M., Kang, S., Wang, X., Pepin, N., and Wu H.: Understanding changes in the water budget driven by climate change in cryospheric-dominated watershed of the northeast Tibetan Plateau, China, Hydrological Processes, 1-19, doi:10.1002/hyp. 13383, 2019.

Please also note the supplement to this comment: https://www.hydrol-earth-syst-sci-discuss.net/hess-2019-302/hess-2019-302-AC1-supplement.pdf
* * *
(a) Past climate (b) Warmer climate

Glacier

Springs and swamps

Streams

Active groundwater layer

Unconsolidated soil layer

Permafrost

Fresh bedrock

**Fig. 1.**

**Supplement:**

[revised manuscript text omitted]

---

## Author Comment (AC2) · 8 Sep 2019

This paper presents the temperature, precipitation and stream variation in the Yangbajain catchment. Interestingly, the estimate the base flow and connect the baseflow variation with the climate change. This is important for the local water resources management and well as for the global groundwater-climate change research. But it should be accepted after a minor revision. Response: Many thanks for the positive comments

and suggestions. We have addressed the reviewer's concerns and suggestions carefully. In the following, we provide point-by-point response to each reviewer comment (blue texts are our responses, while black texts are original comments).

My major comments are: 1. The accuracy of base flow and groundwater storage estimation. As I pointed out in the specific comment, the authors should provide more evidences to show the estimated groundwater storage are correct. Response: Yes, we agree that the results need to be verified by more evidences. However, as we know, at catchment scale, especially in Alpine regions, there are few direct methods to measure water storage at catchment scales, and direct observations of permafrost are even difficult to perform (Lyon et al., 2010; Creutzfeldt et al., 2014; Rogger et al., 2017; Patnaik et al., 2018). Several alternatively indirect methods have been proposed to try to validate the estimation from recession analysis. Vannier et al. (2014) compared the recession analysis based estimation of groundwater storage capacity with the method that estimates storage capacity by multiplying the soil thickness and specific yield. Birkel et al. (2011) used a tracer-constrained process-based conceptual model to validate storage dynamic estimated from the recession analysis based method. These indirect methods are considerable at small catchment with humid climate. Although superconducting gravimetry can measure the storage dynamic directly (Creutzfeldt et al., 2014), it is costly and only available at specific location. Instead, the GRACE data were used to verify our estimations in this study. In addition, we know that groundwater level is rising through recent field investigations. The increases of surface water and shallow groundwater are changing the land cover and NDVI (Figure 1) is rising accordingly in recent years. All these provide evidences to the estimated rising groundwater storage. In fact, not only in the study area but in the whole TP as well as surrounding regions, surface water and groundwater storage are increasing due to climate warming, and hence vegetation conditions are improved (Zhang et al., 2018; Khadka et al., 2018).

2. The explanation on the glacier loss should be deleted. Please see the specific

comment (Line 408-413). Response: it has been revised accordingly.

3. The schematic model (Figure 3). (1) The glacier thickness should increase with the altitude; (2) 'Unconsolidated material' changes 'Unconsolidated soil layer'; (3) Take care of the width of the arrows. Response: Thank you very much for your suggestions. According to your suggestions and those of the second reviewer, we have made corresponding modifications by considering local real situations in the study region, as shown in figure 2.

Specifice comments: Line 115&117 What is the method difference between Lyon et al. (2009) and Kirchner et al. (2009)? And what is the latest advance of the recession analysis? Please clarify. Response: Lyon et al. (2009) method is based on the recession flow analysis developed on the basis of hydraulic groundwater theory by Brutsaert and Nieber (1977) and Brutsaert (2008). However, Kirchner (2009) derived a nonlinear first-order dynamical equations by the conservation-of-mass theory for simulate the streamflow hydrograph from precipitation and evapotranspiration. The power law recession relationship which is used to characterize catchments based on nonlinear reservoir model or a Boussinesq representation of subsurface flow is only a special case in Kirchner's study. In hydrology, the storage-discharge relationship is a fundamental catchment property and can provide a functional form for recession analysis (Lyon et al., 2010; Creutzfeldt et al., 2014). However, to date, there are few direct methods to measure water storage at catchment scales, let alone to measure permafrost change in Alpine regions (e.g., the Qinghai-Tibet Plateau). Thus explicit storage-discharge relationship still remains unknown to us. Creutzfeldt et al. (2014) adopted direct measurements of terrestrial water storage dynamics by means of superconducting gravimetry in a small headwater catchment to derive empirical storage-discharge relationships. As direct measurement remains a major challenge, Birkel et al. (2015) and Soulsby et al. (2015) use a tracer-aided hydrological model to characterize catchment storage. Though many new methods (e.g., tracer-aided model) are proposed, to date, the classical technique of recession flow analysis according to recession flow or flow during no‐rain periods sustained by basin storage (S) is still widely used to provide important information on storage–discharge relationship of the basin (Patnaik et al., 2018). This is because many methods are limited by observations. For instance, in many catchments, especially in Alpine regions, hydrological observations are sparse and direct observations of permafrost are difficult to perform. Most importantly, the recession flow analysis is based on widely available hydrologic data (i.e., streamflow data). As an important component of hydrograph, the nonlinear properties and inconsistency of recession segments among events are emphasized to give better parameterization of recession process through both hydrograph analysis and analytical and numerical simulation (Bogaart et al., 2013; Dralle et al., 2015, 2017; Gao et al., 2017; Hogarth et al., 2014; Roques et al., 2017; Sawaske and Freyberg 2014; Stoelzle et al., 2013). Recession analysis now works as an effective tool to explore catchment-scale physical attributes, such as catchment-scale hydrogeological parameters (saturated hydraulic conductivity, aquifer thickness), active river network dynamic, and storage capacity (Biswal and Kumar, 2014; Pauritsch et al., 2015; Shaw et al., 2016; Troch et al., 2013, Vannier, 2014). The catchment hydrological functions are also revealed through recession analysis. Hydrologic fluxes (actual evaporation, different streamflow components) and state-variables (like storage dynamic) can be estimated from recession analysis (Creutzfeldt et al., 2014; Shaw and Riha, 2012; Szilagyi et al., 2007;). A simple dynamic model can be even developed based recession analysis (Kirchner, 2009; Rusjan and Mikoš, 2015; Teuling et al., 2010). Besides, the streamflow recession patterns are used to unravel the co-evolution of landscape (Bogaart et al., 2016) and also the impact of climate change on permafrost degradation (Lyon et al., 2009; Ploum et al., 2019).

Line 163, 164&168. Please describe the number clearly on the period as well as the hydrologic station. Response: it has been revised accordingly. The air temperature of the Yangbajain Catchment is the areal average value over the whole catchment, which is calculated by the method of meteorological data extrapolation by Prasch et al. (2013). The precipitation and streamflow is the statistical values at the Yangbajain

station.

Line 169-171 How do you get the number of 63% from Fig. 2. And I do not think you can get this number easily only with the data of temperature, precipitation amount and runoff. Response: it is a little bit puzzling. Here we mean the runoff volume in summer account for 63% of the annual streamflow volume and it has been revised accordingly.

Line286-288 The higher grade relational grade is found at the annual scale, how can you say the air temperature also acts a primary role for the base flow? Response: According to the trend analysis of hydro-meteorological factors (e.g., precipitation, Air temperature, etc), we found that baseflow as well as streamflow are both increasing. Through gray relational analysis, we aim to identify the major climatic factors for the increasing streamflow. The result shows that the air temperature compared with precipitation has the higher gray relational grade at annual scale (Table 2). This indicates that the air temperature instead of precipitation acts as a primary factor for the increased streamflow as well as the baseflow. The continuous warming has led to glacier loss and permafrost degradation that contribute to the increasing of streamflow.

Line 339-344 I suggest to shift these sentences above the lines 335-339. Before discussing the trend of the groundwater storage, you should firstly explain the obtained results of groundwater storage are reasonable. I also ask the authors to give more explanation on their obtained groundwater storage, because it does seem consistent between the Grace data and your data. Could the authors give more evidences of the monitored groundwater level? Response: these sentences have been shifted accordingly. As you know in the harsh Yangbajain catchment there are no monitored groundwater wells observed by either official departments or scientific community. At this stage, we have to seek to public data (e.g., GRACE data) for verifications of our estimations. In addition, we know that groundwater level is rising through recent field investigations. The increases of surface water and shallow groundwater are changing the land cover and NDVI (Figure 1) is rising accordingly in recent years. All these provide evidences to the estimated rising groundwater storage. In fact, in the whole TP

as well as surrounding regions, surface water and groundwater storage are increasing due to climate warming, and hence vegetation conditions are improved (Zhang et al., 2018; Khadka et al., 2018)

Line 356-370 I understand the authors try to draw the conclusion 'the increased streamflow is mainly fed by the accelerated glacier retreat rather than frozen ground degradation' through the comparison between four catchments. This is something kind of 'circumstantial evidence'. Could you explain why the frozen ground degradation does not increase the streamflow? Response: Yes, we agree that it is some kind of circumstantial evidence. The frozen ground degradation also contributes to the increasing of streamflow. However, through parallel comparison of different sub-basins (Table 3), we can conclude that the contribution of glacier retreat is much larger than frozen ground degradation. While the mostly significant effects of frozen ground degradation on runoff is that it can increase groundwater storage space and change the behavior of storage-discharge in the catchment. Similar results can be found in many other studies, e.g., Xu et al. (2019), Khadka et al. (2018) and Walvoord and Striegl (2007). For example, Walvoord and Striegl (2007) found that permafrost thawing in an arctic basin has resulted in a general upwards trend in groundwater contribution to streamflow of 0.7-0.9%/yr, however, with no pervasive change in total annual runoff.

Line 408-413. This is quite arbitrary. Although the estimation of glacier loss is reasonable, the loss can be explained in many ways. For example, it could be delivered through the different pathways of shallow aquifer; and it could be exchanged with the aquifers outside the studied region. Sure, it may also infiltrate into the deep fault. But all of these hypotheses need evidences. If you take the one of deep circulation, you should describe clearly the hydrogeologic features of the fault. Is it conductive or not? What is the depth of it? What is the groundwater flow direction inside it? Could you provide the hydrogeologic section map here? If the authors could not provide the discussion above, I suggest the authors to delete this paragraph and leave the glacier loss as an open discussion question here. Response: Yes, we agree with the

reviewer's comments and it has been deleted.

References: Birkel, C., Soulsby, C. and Tetzlaff D.: Conceptual modelling to assess how the interplay of hydrological connectivity, catchment storage and tracer dynamics controls non-stationary water age estimates, Hydrological Processes, 29(13), 2956–2969, doi:10.1002/hyp.10414, 2015. Brutsaert, W.: Long-term groundwater storage trends estimated from streamflow records: Climatic perspective, Water Resources Research, 44(2), 114-125, doi:10.1029/2007WR006518, 2008. Creutzfeldt, B., Troch, P. A., Güntner, A., Ferré, Ty P. A., Graeff, T., and Merz, B.: Storage-discharge relationships at different catchment scales based on local high-precision gravimetry, Hydrological Processes, 28, 1465–1475, 2014. Birkel, C., Soulsby, C., and Tetzlaff, D.: Modelling catchment scale water storage dynamics: Reconciling dynamic storage with tracer inferred passive storage. Hydrological Processes, 25(25), 3924-3936, 2011. Biswal, B., and Nagesh Kumar, D.: Study of dynamic behaviour of recession curves. Hydrological Processes, 28(3), 784-792, 2014. Bogaart, P. W., Rupp, D. E., Selker, J. S., and Van Der Velde, Y.: Late‐time drainage from a sloping Boussinesq aquifer. Water Resources Research, 49(11), 7498-7507, 2013. Bogaart, P. W., Van Der Velde, Y., Lyon, S. W., and Dekker, S. C.: Streamflow recession patterns can help unravel the role of climate and humans in landscape co-evolution. Hydrology and Earth System Sciences, 20(4), 1413-1432, 2016. Brutsaert, W., and Nieber, J. L.: Regionalized drought flow hydrographs from a mature glaciated plateau, Water Resources Research, 13(3), 637-643, 1977. Ding, Y. J., Zhang. S.Q., and Chen, R. S.: Introduction to hydrology in cold regions, Science Press, Beijing, China, 2017. (In Chinese). Dralle, D. N., Karst, N. J., Charalampous, K., Veenstra, A., and Thompson, S. E.: Event-scale power law recession analysis: quantifying methodological uncertainty. Hydrology and Earth System Sciences, 21(1), 65-81, 2017. Dralle, D., Karst, N., and Thompson, S. E.: a, b careful: The challenge of scale invariance for comparative analyses in power law models of the streamflow recession. Geophysical Research Letters, 42(21), 9285-9293, 2015. Gao, M., Chen, X., Liu, J., Zhang, Z., and Cheng, Q. B.: Using Two Parallel Linear Reservoirs to Express Multiple Relations of

Power-Law Recession Curves. Journal of Hydrologic Engineering, 22(7), 04017013, 2017. Hogarth, W. L., Li, L., Lockington, D. A., Stagnitti, F., Parlange, M. B., Barry, D. A., ... and Parlange, J. Y.: Analytical approximation for the recession of a sloping aquifer. Water Resources Research, 50(11), 8564-8570, 2014. Khadka, N., Zhang, G., and Thakuri, S.: Glacial Lakes in the Nepal Himalaya: Inventory and Decadal Dynamics (1977–2017). Remote Sensing, 10, 1913, doi:10.3390/rs10121913, 2018. Kirchner, J.W.: Catchments as simple dynamical systems: catchment characterization, rainfall-runoff modeling, and doing hydrology backward, Water Resources Research, 45, W02429, doi:10.1029/2008WR006912, 2009. Lyon, S. W., Destouni, G., Giesler, R., Humborg, C., Mörth, M., and Seibert, J., et al.: Estimation of permafrost thawing rates in a sub-arctic catchment using recession flow analysis, Hydrology and Earth System Sciences, 13(5), 595-604, 2009. Lyon, S. W., and Destouni, G.: Changes in catchment-scale recession flow properties in response to permafrost thawing in the Yukon River basin, International Journal of Climatology, 30(14), 2138-2145, doi:10.1002/joc.1993, 2010. Pauritsch, M., Birk, S., Wagner, T., Hergarten, S., and Winkler, G.: Analytical approximations of discharge recessions for steeply sloping aquifers in alpine catchments. Water Resources Research, 51(11), 8729-8740, 2015. Patnaik, S., Biswal1, B., Kumar, D. N., Sivakumar, B.: Regional variation of recession flow power‐law exponent, Hydrological Processes, 32, 866–872, 2018. Ploum, S. W., Lyon, S. W., Teuling, A. J., Laudon, H., and van der Velde, Y.: Soil frost effects on streamflow recessions in a subarctic catchment. Hydrological Processes, 33(9), 1304-1316, 2019. Prasch, M., Mauser, W., and Weber, M.: Quantifying present and future glacier melt-water contribution to runoff in a central Himalayan river basin, Cryosphere, 7(3), 889-904, doi:10.5194/tc-7-889-2013, 2013. Rogger, M., Chirico, G. B., Hausmann, H. Krainer, K. Brückl, E. Stadler, P. and Blöschl, G.: Impact of mountain permafrost on flow path and runoff response in a high alpine catchment, Water Resources Research, 53, 1288-1308, doi:10.1002/ 2016WR019341, 2017. Roques, C., Rupp, D. E., and Selker, J. S.: Improved streamflow recession parameter estimation with attention to calculation of$-$ dQ/dt. Advances in water resources, 108,

29-43, 2017. Rusjan, S., and Mikoš, M.: A catchment as a simple dynamical system: Characterization by the streamflow component approach. Journal of Hydrology, 527, 794-808, 2015. Sawaske, S. R., and Freyberg, D. L.: An analysis of trends in baseflow recession and low-flows in rain-dominated coastal streams of the pacific coast. Journal of Hydrology, 519, 599-610, 2014. Shaw, S. B.: Investigating the linkage between streamflow recession rates and channel network contraction in a mesoscale catchment in New York state. Hydrological Processes, 30(3), 479-492, 2016. Shaw, S. B., and Riha, S. J.: Examining individual recession events instead of a data cloud: Using a modified interpretation of dQ/dt–Q streamflow recession in glaciated watersheds to better inform models of low flow. Journal of hydrology, 434, 46-54, 2012. Soulsby, C., Birkel, C., Geris, J., Dick, J., Tunaley, C. and Tetzlaff, D.: Stream water age distributions controlled by storage dynamics and nonlinear hydrologic connectivity: Modeling with high-resolution isotope data, Water Resources Research, 51, 7759–7776, doi:10.1002/2015WR017888, 2015. Stoelzle, M., Stahl, K., and Weiler, M.: Are streamflow recession characteristics really characteristic?. Hydrology and Earth System Sciences, 17(2), 817-828, 2013. Szilagyi, J., Gribovszki, Z., and Kalicz, P: Estimation of catchment-scale evapotranspiration from baseflow recession data: Numerical model and practical application results. Journal of hydrology, 336(1-2), 206-217, 2007. Teuling, A. J., Lehner, I., Kirchner, J. W., and Seneviratne, S. I.: Catchments as simple dynamical systems: Experience from a Swiss prealpine catchment. Water Resources Research, 46(10), 2010. Troch, P. A., Berne, A., Bogaart, P., Harman, C., Hilberts, A. G., Lyon, S. W., ... and Teuling, A. J.: The importance of hydraulic groundwater theory in catchment hydrology: The legacy of Wilfried Brutsaert and Jean‐Yves Parlange. Water Resources Research, 49(9), 5099-5116, 2013. Vannier, O., Braud, I., and Anquetin, S.: Regional estimation of catchment‐scale soil properties by means of streamflow recession analysis for use in distributed hydrological models. Hydrological Processes, 28(26), 6276-6291, 2014. Walvoord, M. A., and Striegl, R. G.: Increased groundwater to stream discharge from permafrost thawing in the Yukon River basin: Potential

impacts on lateral export of carbon and nitrogen, Geophysical Research Letters, 34(12), 123-134, doi:10.1029/2007GL030216, 2007. Xu, M., Kang, S., Wang, X., Pepin, N., and Wu H.: Understanding changes in the water budget driven by climate change in cryospheric-dominated watershed of the northeast Tibetan Plateau, China, Hydrological Processes, 1-19, doi:10.1002/hyp. 13383, 2019. Zhang, Z. X., Chang, J., and Xu, C. Y., et al.: The response of lake area and vegetation cover variations to climate change over the Qinghai-Tibetan Plateau during the past 30 years, Science of the Total Environment, 635, 443-451, 2018.

Please also note the supplement to this comment:
https://www.hydrol-earth-syst-sci-discuss.net/hess-2019-302/hess-2019-302-AC2-supplement.pdf
* * *

**Fig. 1.**

[Figure]

(a) Past climate

(b) Warmer climate

| | |
|---|---|
| Glacier | |
| Springs and swamps | |
| Streams | |
| Active groundwater layer | |
| Unconsolidated soil layer | |
| Permafrost | |
| Fresh bedrock | |

**Fig. 2.**

[Figure]

---

## Author Response (AR2)

Dear Editor and Reviewers,

The manuscript entitled "Understanding the effects of climate warming on streamflow and active groundwater storage in an alpine catchment, upper Lhasa River" has been thoroughly revised according to the anonymous reviewers' comments. Major revisions have been made for improving its quality. For example, we have elaborated in detail the mechanism of hydrological processes. Figure 3 was re-draw according to the real situations of topography and distribution of glaciers and permafrost in the Yangbajain catchment according to the reviewer's suggestions. And we added the GRACE as well as NDVI data to verify our estimations in this study to provide an evidence for the estimated increasing water storage.

For any further corrections and requirements, the authors are ready here for your critiques.

Correspondence and phone calls about the paper should be directed to Prof. Liu Jintao at the following address, phone and fax number, and e-mail address: State Key Laboratory of Hydrology-Water Resources and Hydraulic Engineering, Hohai University; Adress: 1 Xikang Road, Nanjing 210098, People's Republic of China; Tel:

+86-25-83787803; Fax: +86-025-83786606; E-mail:jtliu@hhu.edu.cn.

Thanks very much for your attentions to our paper again.

Sincerely yours,

Liu Jintao

**Editor Decision: Publish subject to revisions (further review by editor and referees)** (08 Nov 2019) by Fuqiang Tian

Comments to the Author:

Dear Authors,

I got the comments from two Referees. I understand the field experimental data in Tibetan Plateau is really rare and valuable. I suggest the authors to address the following comments (repeated from the reviewer's comment):

**Response:** Many thanks for your kindly helps. We have address all the following comments.

1) To elaborate the mechanisms of hydrological processes, hydrological cycle and the relations between recharge and drainage in alpine region.

**Response:** We elaborated in detail this issue again and added many newly studies to try to explain it clearly. See details in **Revised Lines [59-61, 70-77]**.

2) To elaborate the main source of streamflow.

**Response:** According to your suggestions, we have added all of them to elaborate it fully and clearly. First, according to our data and the results of Investigation of River and Lake (by Guan et al., 1984) in the first Tibetan Plateau Scientific Expedition and Research, the runoff is mainly recharged by rainfall. For instance, the catchment has a summer (June-August) monsoon with 73% of the yearly precipitation, summer streamflow recharged mainly by monsoon rainfall and meltwater accounts for approximately 63% of the yearly streamflow. According to Guan et al.'s results, 48% of total runoff is fed by rainfall. Revised Lines [157-157, 342-346]

Then we revealed that air temperature acts as a primary factor for the increased streamflow as well as the baseflow through gray relational analysis. **Revised Lines [261-265]**. Furthermore, through parallel comparisons with other sub-basins in the Lhasa River Basin we can indirectly reveal that the increased streamflow at the Yangbajain station is mainly fed by the accelerated glacier retreat.

However, without applications of hydrologic models and hydrological isotopes, the accurate number or ratio to quantify the contribution of glaciers meltwater and permafrost degradation to streamflow have not been provide in this study, and we suggested to do it in future study in the final part of conclusions **Lines [405-409]**.

3) Revise Figure 3.

**Response:** it has been revised accordingly.

4) Try to use more data in this paper in available.

**Response:** it has been added accordingly.

Suggestions for revision or reasons for rejection (will be published if the paper is accepted for final publication)

Journal: HESS

Title: Quantifying streamflow and active groundwater storage in response to climate warming in an alpine catchment, upper Lhasa River

MS No.:HESS_2019_302

I have gone through the article several times trying to establish how it can be salvaged as the research done is significant and scientific. However, the authors still did not fully address the issues pointed out by the reviewers.

**Response:** Many thanks for your kindly helps. We have fully addressed your concerned issues in the revised version. In the following, point-to-point responses will be given.

1.The comment "the mechanism of hydrological process, hydrological cycle and the relationship between recharge and drainage of water in alpine region are not described in detail" Unfortunately, the authors did not address this key issue. The mechanism of hydrological process, hydrological cycle and the relationship between recharge and drainage of water in alpine region is the theoretical basis. So authors MUST add it.

**Response:** thank you for your kindly reminders. We elaborated in detail this issue again and added many newly studies to try to explain it clearly. See details in **Revised Lines [59-61, 70-77].**

2.Figure 3, the color is not clear, especially Glacier and Active groundwater layer, spring and swamps and permafrost. Are the names of the Active groundwater layer and unconsolidated soil layer right? This is can be done by Wang et al. (2018). Wang, W., Wu, T., Zhao, L., Li, R., Zhu, X., Wang, W., ... & Hao, J. (2018). Exploring the ground ice recharge near permafrost table on the central Qinghai-Tibet Plateau using chemical and isotopic data. Journal of Hydrology, 560, 220-229.

**Response:** Thank you very much for your suggestions. According to your suggestions, we have made corresponding modifications by considering local real situations in the study area. We deleted the unconsolidated soil layer and changed "Active groundwater layer" to "Active layer" by referring to the papers of Wang et al. (2018) and Li et al. (2018).

3.The comment "This work has been found that the increased streamflow is mainly fed by the accelerated glacier retreat due to climate warming. There are many factors for the increase of streamflow. The accelerated glacier is just one of all factors. For example, the increase of precipitation, the degradation of frozen soil, the melting of underground ice, and the supply of supra-permafrost water. So I suggest that authors first figure out what is the main sources of streamflow in the study area? Then analyzed the contribution of the recharge sources to runoff based on the variation of all factors under the climate warming. Finally, the main reason for the increase for runoff is obtained." Unfortunately, the authors did not address this key issue. This issue is critical, especially for this study. So authors MUST add it.

**Response:** yes, we have ignored these important issues in the ms. According to your suggestions, we have added all of them to elaborate it fully and clearly. First, according to our data and the results of Investigation of River and Lake (by Guan et al., 1984) in

the First Tibetan Plateau Scientific Expedition and Research, the runoff is mainly recharged by rainfall. For instance, the catchment has a summer (June-August) monsoon with 73% of the yearly precipitation, summer streamflow recharged mainly by monsoon rainfall and meltwater accounts for approximately 63% of the yearly streamflow. According to Guan et al.'s results, 48% of total runoff is fed by rainfall. **Revised Lines [157-157, 342-346]**

Then we revealed that air temperature acts as a primary factor for the increased streamflow as well as the baseflow through gray relational analysis. **Revised Lines [261-265]**. Furthermore, through parallel comparisons with other sub-basins in the Lhasa River Basin we can indirectly reveal that the increased streamflow at the Yangbajain station is mainly fed by the accelerated glacier retreat.

However, without applications of hydrologic models and hydrological isotopes, the accurate number or ratio to quantify the contribution of glaciers meltwater and permafrost degradation to streamflow have not been provide in this study, and we suggested to do it in future study in the final part of conclusions **Lines [405-409]**.

4.I think the authors may need more data to support their arguments and the discussion needs to be strengthened.

**Response:** In the revised ms, the GRACE as well as the NDVI data were used to verify our estimations in this study. In addition, we know that groundwater level is rising through recent field investigations. However, we don't have continuous data of groundwater stages. So the increases of surface water and shallow groundwater are changing the land cover and NDVI (Figure 10) is rising accordingly in recent years. All these provide evidences to the estimated rising groundwater storage. Many parts of results and discussions have been revised. See in **Revised Lines [318-322; 342-346; 352-354].**

**References:**

Guan, Z. H., Chen, C. Y., Kuang, Y. X., Fan Y. Q., Zhang, Y. S., and Chen, Z. M. et al.: Rivers and Lakes in Tibetan. Rivers and lakes in Tibet. Beijing: Science and Technology Press, 1984 (in Chinese).

Li, Z. J., Li, Z. X., Song, L. L., Ma, J. Z., and Song Y.: Environment significance and hydrochemical characteristics of suprapermafrost water in the source region of the Yangtze River, Science of the Total Environment, 644, 1141-1151, 2018.

Wang, W. F., Wu, T. H., Zhao, L., Li R., Zhu X. F., Wang, W. R., Yang, S. H., Qin, Y. H., and Hao, J. M.: Exploring the ground ice recharge near permafrost table on the central Qinghai-Tibet Plateau using chemical and isotopic data, Journal of Hydrology, 560, 220-229, 2018.

Wang, Y. H., Yang, H. B., Gao, B., Wang, T. H., Qin, Y., and Yang, D. W.: Frozen ground degradation may reduce future runoff in the headwaters of an inland river on the northeastern Tibetan Plateau, Journal of Hydrology, 564, 1153-1164, 2018.